# Health care seeking behavior for common childhood illnesses in Birendranagar municipality, Surkhet, Nepal: 2018

Ganga Tiwari[1]*, Ajoy Kumar Thakur[1], Sushil Pokhrel[2], Ganesh Tiwari[3], Durga Prasad Pahari[4]

1 Department of Community Medicine and Public Health, Maharajgunj Medical Campus, Institute of Medicine, Tribhuvan University, Kathmandu, Nepal, 2 Department of Orthopedics and Trauma Surgery, Maharajgunj Medical Campus, Institute of Medicine, Tribhuvan University, Kathmandu, Nepal, 3 Department of Medicine, Karnali Academy of Health Sciences, Jumla, Nepal, 4 Central Department of Public Health, Institute of Medicine, Tribhuvan University, Kathmandu, Nepal

* tiwariganga99@gmail.com

## Abstract

**Data Availability Statement:** All relevant data are within the manuscript and its Supporting information files.

### Background

Appropriate and prompt health care seeking by parents or caretakers can reduce morbidity and mortality among under-five year children. Although remarkable progress has been made in the reduction of under-five mortality, still the under-five mortality rate is high in Nepal. There are few studies on health care seeking behavior among caretakers in Nepal. Therefore, this study was conducted to determine the prevailing health care seeking behavior of caretakers on common childhood illness of under five year children and to identify the association of socio demographic, economic, illness related and health system related factors with health care seeking behavior in 2018.

### Methods

A community based descriptive cross-sectional study was conducted from September to November 2018. Data were collected using a pretested semi-structured interview schedule. Both descriptive and inferential statistics were used to present the data. Bivariate and multivariate logistic regression analysis was used to identify the factors associated with health care seeking behavior.

### Results

A total of 387 caretakers participated in the study. Of these, 84.8% sought any type of care and 15.2% did nothing. Amongst those who sought care 42.4% visited the pharmacy directly, 25.3% visited the health facility. Amongst those who visited a health facility, 37.2% of caretakers sought prompt health care. Common danger sign stated by caretakers was fever in children (92.4%). Secondary education(AOR = 0.357, 95%CI = 0.142–0.896), involvement in service as an occupation(AOR = 3.533, 95%CI = 1.096–11.384), distance to reach nearest health facility(0.957, 95%CI = 0.923–0.993) and perceived severity of illness;

**Funding:** The authors receive no specific findings for this work.

**Competing interests:** Authors have no competing interest.

**Abbreviations:** AOR, Adjusted Odds Ratio; ARI, Acute Respiratory Tract Infection; CB-IMNCI, Community Based Integrated Management of Neonatal and Childhood Illness; CI, Confidence Interval; COR, Crude Odds Ratio; FCHVs, Female Community Health Volunteers; IMCI, Integrated Management of Childhood Illness; IOM, Institute of Medicine; IRC, Institutional Review Committee; MPH, Master in Public Health; NDHS, Nepal Demographic Health Survey; NRs, Nepalese Rupees; SPSS, Statistical Package for the Social Sciences; TU, Tribhuvan University; UNICEF, United Nations International Children's Fund; WHO, World Health Organization.

moderate severity (7.612, 95%CI = 2.127–27.242), severe severity (AOR = 15.563, 95%CI = 3.495–69.308) were found to be significantly associated with health care seeking behavior.

## Conclusion

Strong policies and regulations should be formulated and implemented at Birendranagar municipality of Surkhet district to prevent direct purchase of medicines from pharmacies without any consultation. It is essential to conduct the health awareness program at community level on early recognition of danger signs and importance of consulting health facilities.

## Introduction

Health or care-seeking behavior has been defined as any action undertaken by individuals who perceive themselves to have a health problem or to be ill for the purpose of finding an appropriate remedy [1].

Illnesses such as diarrhea, pneumonia, measles, malaria, and malnutrition remain major contributors to mortality among under-five children globally. Poor and delayed health care seeking has contributed to 70% of all deaths among under-five children [2].

Globally 5.5 million of under-five year children died in 2017, under-five mortality rate was 39 per thousand live births [3]. Although remarkable success has been made in survival of children since 1990, still under-five mortality rate is high in Sub-Saharan Africa, Central and Southern Asia, which accounts for more than 80 percent of under-five deaths in 2018 [4]. In Nepal, the under-five mortality rate is 39 deaths per 1,000 live births [5].

The prevalence of health care seeking is low in developing countries as compared to the developed countries. Globally around 78% of children with symptoms of Acute Respiratory Infections (ARI) were taken to the health care provider, but the coverage is only 43% in low-income countries. As many children are not taken for treatment in low-income countries disease management and surveillance has been a difficult process [6].

It has been suggested from the studies that timely health care seeking by the caretakers or family members could prevent morbidity and mortality of under-five children [7, 8]. In Nepal, there are few published studies on healthcare-seeking behavior of caretakers, and no studies have been done in the Surkhet district so far.

Therefore, this study was conducted to determine the prevailing health care seeking behavior of caretakers on common childhood illness of under five year children and to identify the association of socio demographic, economic, illness related and health system related factors with health care seeking behavior in 2018.

## Materials and methods

### Study design

A Community based descriptive cross-sectional study was carried at Birendranagar Municipality of Surkhet district.

### Study duration and area

The study was carried from September 2018 to November 2018. Surkhet district is located about 600 Kilometer west of Kathmandu. The district's area is 2,451 square kilometers. There

are five municipalities and four rural municipalities. The study was conducted in Birendranagar municipality. The total population of Birendranagar municipality is 100,458. The under-five population is 11,787 and total households are 23, 715 [9].

There are 2 government hospitals, 3 primary health care centers, 47 health posts, 150 primary health care outreach clinics, 184 immunization clinic, 987 Female Community Health Volunteers (FCHVs), and many private hospitals and pharmacies in Surkhet district [10].

## Sample size determination and sampling procedures

The sample size was calculated by taking the prevalence of healthcare-seeking behavior p = 81.4% from the study conducted in Lalitpur, Nepal [11]. Single population proportion formula; $n = z^2pq/d^2$ was used for sample size calculation [12], where z = 1.96 at 95% confidence interval, the margin of error (d) = 5%, non-response rate = 10%, As multistage random sampling method was used, to minimize the sampling error the obtained sample size was multiplied by design effect. Design effect 1.5 was used considering a previous study [13]. Hence, the total sample size taken was 387.

There were a total of sixteen wards in Birendranagar municipality. Out of 16 wards, 9 wards were selected randomly using the lottery method. A list of all caretakers having under-five year children meeting the inclusion criteria was made with the help of Female Community Health Volunteers (FCHVs) and vitamin A register of the respective wards. A systematic random sampling technique was used to obtain the required sample size (Fig 1).

## Tool and techniques of data collection

Pretested, semi-structured interview schedule was used for the data collection. Kuppuswamy's socioeconomic status scale modified in the context of Nepal was used to the measure socioeconomic status [14]. Questions related to health care seeking behaviors were adopted from the United Nations Children's Fund Integrated Management of Childhood Illness (UNICEF's IMCI) household-level questionnaire for under-five year children [15] and reviewing other relevant published studies on a similar topic. The interview schedule was translated in the Nepali language and it was pretested among 10% caretakers, those caretakers who were involved in pretesting were not included in the final study. Face to face interview technique was used.

## Data processing and analysis

Data checking and editing were done manually. Coding and data entry was done in EpiData 3.1 version. Data were exported and analyzed in Statistical Package for the Social Sciences (SPSS) version 21. In descriptive statistics; frequency, percentage, mean and standard deviation were used. In inferential statistics chi-square test was used to identify the association between the outcome variable and independent variables. Variables having p-value $\leq$ 0.2 in bivariate analysis were entered in multivariate analysis taking the reference of various published studies [16–19]. Multicollinearity among the selected independent variables was checked through the variance inflation factor (VIF), and there were no multicollinearity issues among those variables. p-value <0.05 in multivariate analysis was used to declare that there was a statistical association.

## Study variables

The conceptual framework (Fig 2) is based on the Anderson Health Care Utilization Model [14]. This model is a conceptual model aimed at demonstrating the factors that lead to the utilization of health care services.

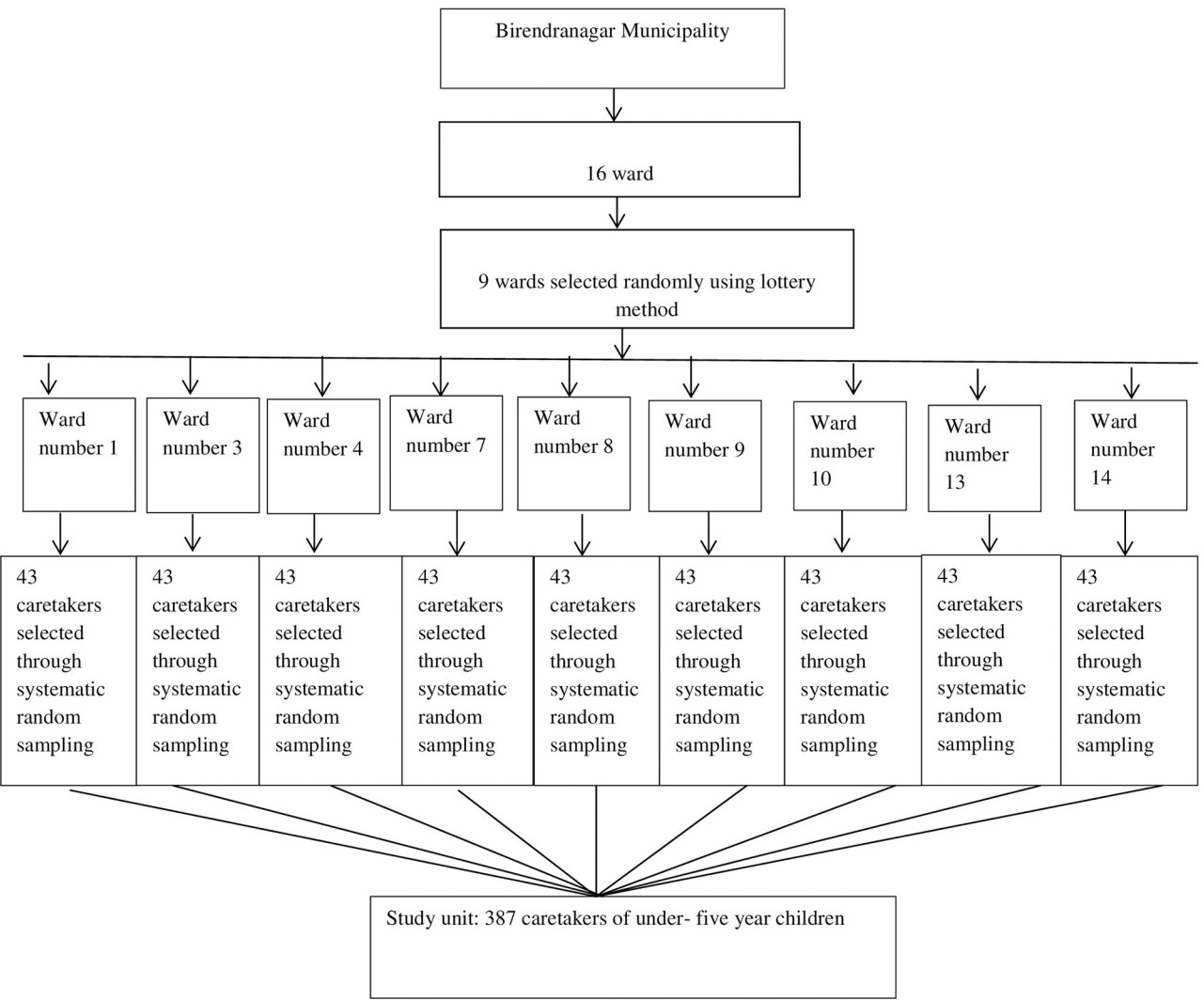

**Fig 1. Sampling technique.**

## Operational definitions

Health Care Seeking Behavior was categorized into appropriate health care seeking behavior and inappropriate healthcare-seeking behavior.

**Appropriate health care seeking behavior.** Health care-seeking from health facilities such as hospitals, nursing homes, health centers, clinics, health posts, etc. during the illness of under five-year children was categorized as appropriate healthcare-seeking behavior.

**Inappropriate health care seeking behavior.** Consultation of pharmacists for medical care, self-purchase of medicine without a prescription, using home remedies, visiting traditional healers, and not seeking any care for during the illness of under-five children were classified as inappropriate healthcare-seeking behavior.

**Prompt health care seeking behavior.** Health care-seeking from the health facilities within 24 hours of recognition of child's illness.

**Delayed health care seeking behavior.** Health care-seeking from the health facilities after 24 hours of recognition of child's illness.

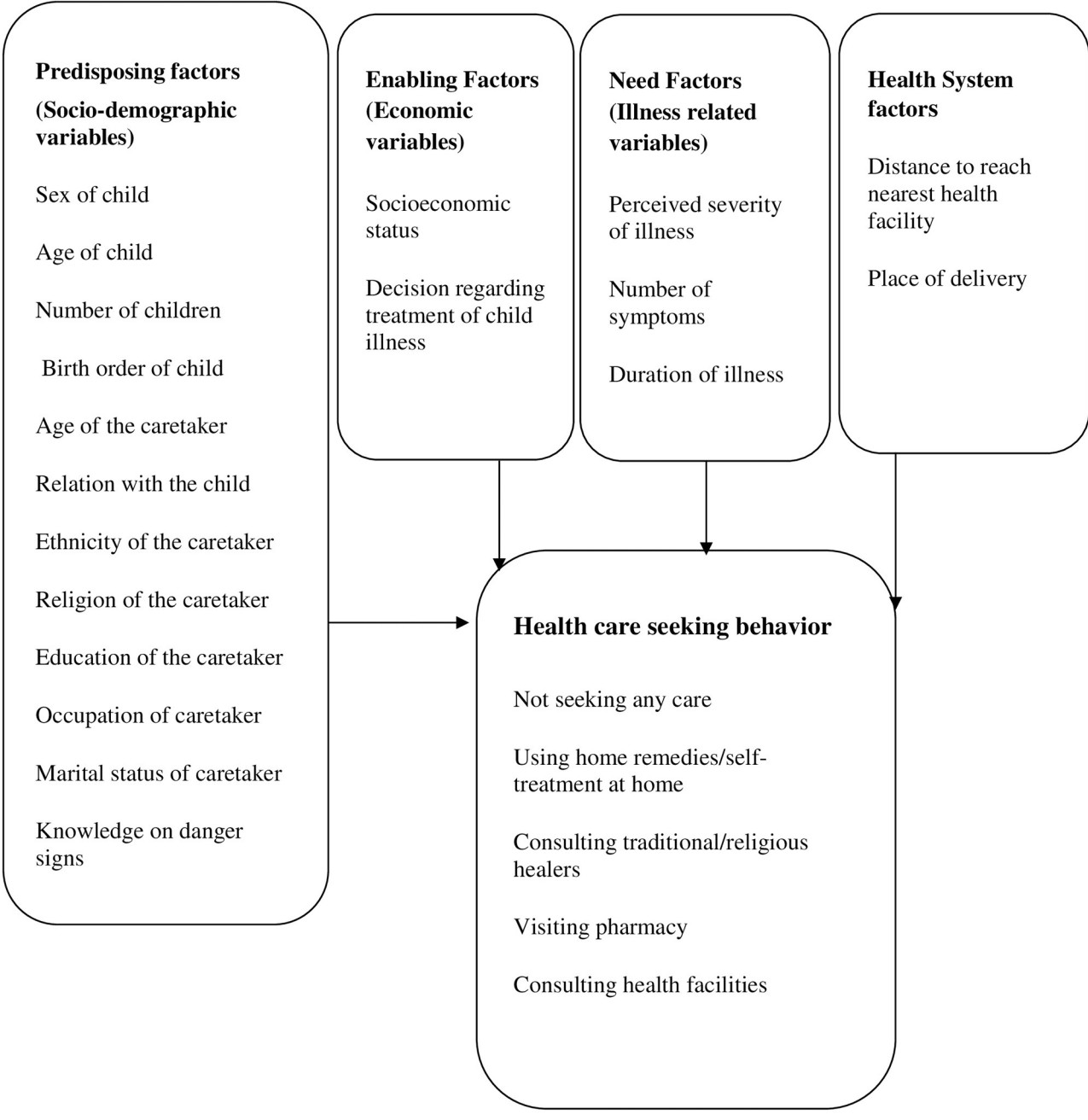

**Fig 2. Conceptual framework.**

**Primary caretakers.** Any adult mainly female but can be male also who is responsible for the routine care of the under-five year child. Primary caretakers include mother, father, grandmother, grandfather, and aunts.

**Common childhood illnesses.** In this study, common childhood illnesses are Acute Respiratory Infections (ARI), diarrhea, and fever perceived by caretakers.

**Acute respiratory infection.** ARI is a cough accompanied by difficulty in breathing as perceived by caretakers in their under-five-year children for less than two weeks at any time within the one-month duration from the day of the interview.

**History of difficulty in breathing.** History of difficulty in breathing is defined as the presence of difficulty in breathing among under-five children which includes fast breathing, different breath sounds like wheezing and stridor, or chest in-drawing perceived by caretakers at any time within the one-month duration from the day of the interview.

**Diarrhea.** If the caretaker described that their sick children had three or more than three loose stools per day at any time within one month duration form the day of the interview.

**Fever.** Caretaker's subjective evaluation fever or hot body temperature in children.

*Perceived severity of illness.* Perceived severity was based on the subjective evaluation of illness by caretakers on the basis of discomfort present in child and it was categorized as mild, moderate and severe.

**Knowledge of the caretaker regarding danger signs of the under-five year children.** Knowledge of the caretaker was measured in terms of the number of danger signs as stated by the caretakers.

**Ethical approval and informed consent.** Ethical approval was obtained from the Institutional Review Committee (IRC) of the Institute of Medicine, Tribhuvan University. Permission was taken from concerned authority (Public Health Service Office Surkhet and Birendranagar Municipality Office Surkhet). The objective of the study was explained to the caretakers; both verbal and written consent was obtained before the interview. Confidentiality and anonymity were maintained.

**Inclusion criteria.** Caretakers of under-five children residing in the Birendranagar municipality for more than six months duration and having the children with a recent episode of illness in the past one-month duration were included in the study.

## Result

### Socio-demographic characteristics of caretakers and under-five children

Table 1 shows the socio-demographic characteristics of the caretakers. The mean age of the caretakers was 29.35±10.03 years. Most of them (82.2%) were mothers. Regarding the occupation more than half (59.4%) of the caretakers were homemakers. About one third (31.5%) of the respondents had obtained secondary education. The mean age of the under-five children was 29.54±16.13 months.

### Different health care seeking behavior

Table 2 shows the health care seeking behavior of caretakers. Majority (84.8%) of the caretakers sought care from different sources. The most common (42.4%) health care seeking behavior was visiting pharmacy directly without any consultation (Fig 3). Among the caretakers who visited the pharmacy, majority (89.3%) stated that fast and easy access of medicine at pharmacy.

### Preferred type of health facilities by caretakers

Table 3 presents the type of health facility preferred by caretakers for the treatment of childhood illness. More than half (61.4%) of the caretakers went to private health facility followed by (38.6%) of caretakers who went to government health facility. Regarding the duration of seeking treatment more than half (62.7%) of the caretakers sought health care after 24 hours of the onset of illness while (37.2%) sought prompt health care within 24 hours of the onset of illness (Fig 4).

**Table 1. Socio-demographic characteristics of the caretakers and children n = 387.**

| Characteristics | Response | Number | Percentage |
|---|---|---|---|
| **Age of the caretakers** | Less than 20 years | 23 | 5.9 |
| | 20 to 30 years | 220 | 56.8 |
| | More than 30 years | 144 | 37.2 |
| Mean ± SD (29.35±10.03 years) | | | |
| **Ethnicity of the caretakers** | Brahmin | 103 | 26.6 |
| | Chhetri | 99 | 25.6 |
| | Aadibashi/Janajati | 64 | 16.5 |
| | Dalit | 68 | 17.6 |
| | Thakuri/Sanyashi | 30 | 7.8 |
| | Muslim | 23 | 5.9 |
| **Occupation of the caretakers** | Homemakers | 230 | 59.4 |
| | Agriculture | 48 | 12.4 |
| | Business | 47 | 12.1 |
| | Service | 27 | 7 |
| | Students | 18 | 4.7 |
| | Labor | 17 | 4.4 |
| **Educational status of the caretakers** | Illiterate | 24 | 6.2 |
| | Informal or just literate | 59 | 15.2 |
| | Primary | 68 | 17.6 |
| | Secondary | 122 | 31.5 |
| | Higher secondary | 71 | 18.3 |
| | Graduate post-graduate or above | 43 | 11.1 |
| **Socio-economic status** | Upper and upper middle class | 135 | 34.9 |
| | Middle class | 129 | 33.3 |
| | Lower class | 123 | 31.8 |
| **Type of Family** | Nuclear | 254 | 65.6 |
| | Joint | 133 | 34.4 |
| **Relation with child** | Mother | 318 | 82.2 |
| | Father | 15 | 3.9 |
| | Grandmother | 36 | 9.3 |
| | Aunty or other relatives | 14 | 3.6 |
| | Grandfather | 4 | 1 |
| **Age of the children** | Less than 12 months | 69 | 17.8 |
| | More than 12 months | 318 | 82.2 |
| Mean ± SD (29.54±16.130 months) | | | |
| **Sex of the children** | Male | 224 | 57.9 |
| | Female | 163 | 42.1 |
| **Number of children in the family** | ≤2 | 310 | 80.1 |
| | 3–4 | 70 | 18.1 |
| | >4 | 7 | 1.8 |

## Caretaker's knowledge on danger signs of childhood illness

Table 4 presents the caretaker's knowledge on danger signs. Less than three fourth (71.1%) had heard about at least one of the danger signs. The most common (92.4%) danger sign stated was fever in child.

**Table 2. Distribution of respondents according to the reasons for different health care seeking behavior.**

| Characteristics | | Number | Percentage |
|---|---|---|---|
| **Care sought for childhood illness or not** | Yes | 328 | 84.8 |
| | No | 59 | 15.2 |
| **Reasons for consulting pharmacy directly (n = 139)**\*\* | | | |
| Fast and easy access of medicine at pharmacy | | 109 | 89.3 |
| Pharmacy was nearby | | 84 | 68.9 |
| Don't have to wait in line | | 79 | 64.8 |
| It is costlier to consult a doctor | | 18 | 14.8 |
| Illness was mild | | 13 | 10.7 |
| **Reasons for visiting health facility (n = 83)** \*\* | | | |
| To avoid complications | | 64 | 78 |
| Good treatment is available | | 61 | 74.4 |
| Illness became severe | | 42 | 51.2 |
| Health facility is nearby | | 10 | 12.2 |
| **Reasons for consulting traditional healer (n = 57)**\*\* | | | |
| Illness was caused by an evil spirit | | 43 | 75.4 |
| Because of repeated illness | | 19 | 33.3 |
| Less costly | | 13 | 22.8 |
| Traditional healer is nearby | | 11 | 19.3 |
| Medical care was not effective | | 4 | 7 |
| **Reasons for using home remedies (n = 49)**\*\* | | | |
| Due to mild illness | | 48 | 98 |
| Less costly | | 22 | 44.9 |
| Due to cultural beliefs | | 11 | 22.4 |
| Home remedies do not cause any side effects/harm | | 9 | 18.4 |
| **Reasons for not seeking any care (n = 59)** \*\* | | | |
| Illness was mild so waited for self- recovery | | 52 | 94.5 |
| Treatment in health facility is costly | | 25 | 45.5 |
| Busy at work, could not get time to go to the health facility | | 20 | 36.4 |
| Long waiting time at health facility | | 8 | 14.5 |
| Health facility is at far distance | | 7 | 12.7 |

## Factors associated with health care seeking behavior of caretakers for their under-five children's illness

Table 5 reveals about the factors associated with health care seeking behavior; twelve variables (ethnicity, educational status, occupation, socioeconomic status, distance to reach the nearest health facility, place of delivery, perceived severity, history of difficulty breathing, knowledge of danger signs, number of symptoms, duration of illness, number of children in a family member) that exhibited significant association with health care seeking in bivariate analysis i.e. p-value $\leq 0.2$ were further analyzed in multivariate analysis using binomial logistic regression. Crude Odds Ratio (COR) was calculated by bivariate analysis in binomial logistic regression. Adjusted Odds Ratio (AOR) was calculated by multivariate analysis in binomial logistic regression. Multivariate analysis was done for the adjustment of possible confounders. Hosmer Lemeshow test, the goodness of fit of the model was assessed. The test statistic was 0.500 ($>0.05$) that showed that the model adequately fits the data.

In multivariate regression analysis, caretakers with secondary education were less likely (AOR = 0.3, 95% CI = 0.1–0.8) to seek appropriate healthcare-seeking behavior than those

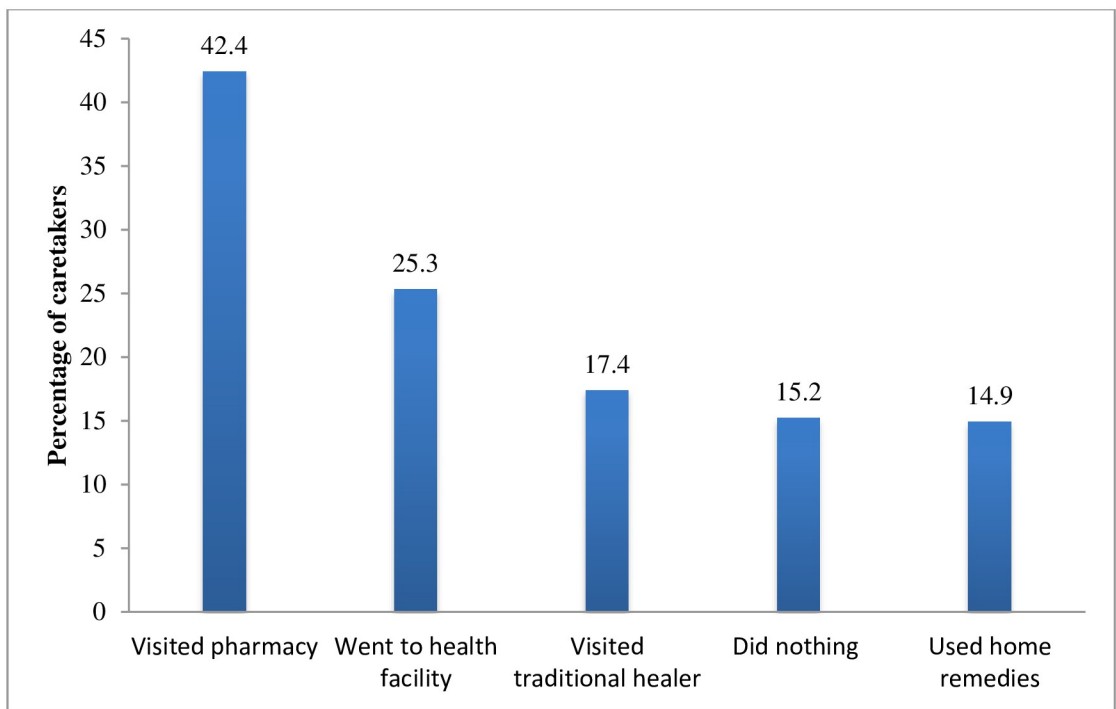

**Fig 3. Health care-seeking behaviors of caretakers.**

who had no schooling or had gained primary education, and higher educational status. Those caretakers who were involved in service as occupation were three times (AOR = 3.5, 95% CI = 1.0–11.3) more likely to seek appropriate health-seeking than those who were involved in other occupations. Similarly, with the increasing distance to reach the nearest health facility,

**Table 3. Distribution of respondents by preferred type of health facilities.**

| Characteristics | Number | Percentage |
|---|---|---|
| **Preferred health facility** | | |
| Private health facility | 51 | 61.4 |
| Government health facility | 32 | 38.6 |
| **Reasons for going to the private health facility (n = 51)**\*\* | | |
| Less waiting time | 43 | 84.3 |
| Good treatment | 42 | 82.4 |
| Private health facility is nearby | 14 | 27.5 |
| Health workers are available | 20 | 39.4 |
| Good behavior of health worker | 14 | 27.5 |
| **Reasons going to the government health facility (n = 32)** \*\* | | |
| Good treatment | 25 | 78.1 |
| Less costly | 21 | 65.6 |
| Government health facility is nearby | 14 | 43.8 |
| Insurance provision | 3 | 9.4 |

Multiple responses\*\*.

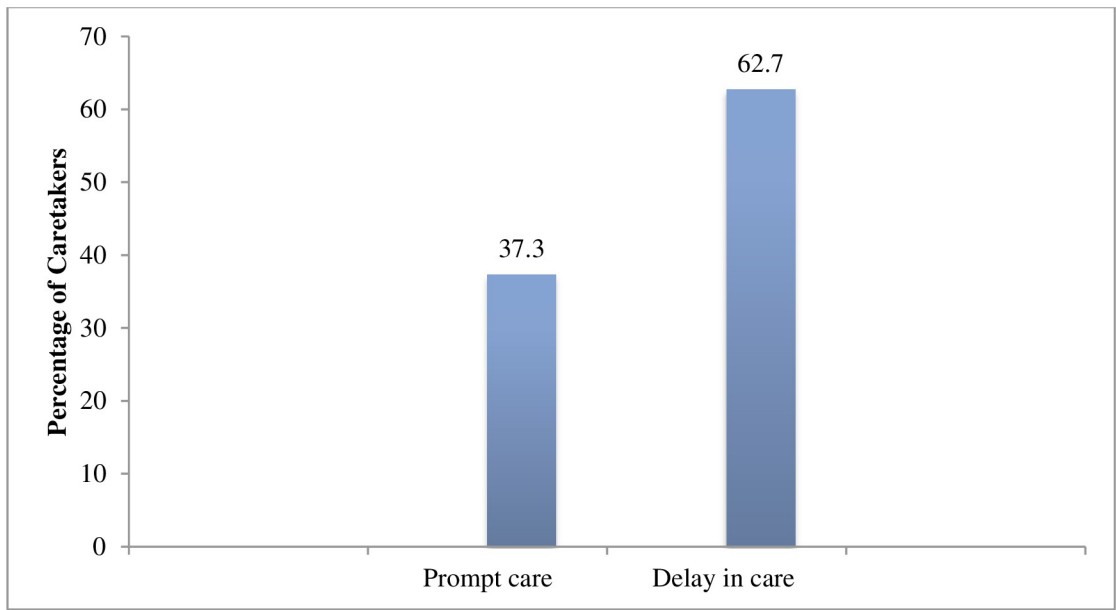

**Fig 4. Duration of health care seeking from the health facility.**

caretakers were less likely to seek appropriate health seeking behavior (AOR = 0.9, 95% CI = 0.9–0.9). Regarding perceived severity those caretakers who perceived that their children had moderate severity of illness were around eight times (AOR = 7.6, 95% CI = 2.1–27.2) more likely to seek appropriate health-seeking behavior, similarly, those caretakers who perceived that their children had a severe illness were fifteen times (AOR = 15.5, 95% CI = 3.4–69.3) more likely to seek appropriate health care than those who perceived illness as mild.

**Table 4. Distribution of caretakers according to knowledge on danger signs of childhood illness.**

| Characteristics | Number | Percentage |
|---|---|---|
| **Heard about danger signs (n = 387)** | | |
| Yes | 275 | 71.1 |
| No | 112 | 28.9 |
| **Danger signs** ** | | |
| Child develops fever | 254 | 92.4 |
| Child becomes sicker | 231 | 84.0 |
| Child has difficulty in breathing | 214 | 77.8 |
| Child has fast breathing | 200 | 72.7 |
| Child is unable to drink or breastfeed | 101 | 36.7 |
| Child becomes unconscious | 90 | 32.7 |
| Child drinks poorly | 87 | 31.6 |
| Child vomits everything | 86 | 31.3 |
| Child has blood in the stool | 75 | 27.3 |
| Child has Convulsions | 17 | 6.2 |

** Multiple responses.

**Table 5. Bivariate and multivariate association of different variables with health care seeking behavior.**

| Characteristics | | Health care Seeking Behavior | | COR (95% CI) | AOR (95% CI) | P-value |
|---|---|---|---|---|---|---|
| | | Appropriate n(%) | Inappropriate n(%) | | | |
| **Ethnicity** | Relatively disadvantage (Ref) | 10 (11.0) | 81 (89.0) | 1 | 1 | |
| | Relatively advantaged | 73 (24.7) | 223 (75.3) | 2.6 (1.3–5.3) | 1.6 (0.6–3.9) | 0.309 |
| **Educational status of the caretakers** | No schooling or up to primary education (Ref) | 26 (17.2) | 125 (82.8) | 1 | 1 | |
| | Secondary education | 15 (12.3) | 107 (87.7) | 0.2 (0.6–0.3) | 0.35 (0.1–0.8) | 0.028* |
| | College or University education | 42 (36.8) | 72 (63.2) | 2.8 (1.5–4.9) | 1.36 (0.6–3.9) | 0.453 |
| **Occupation of caretaker** | Other than service (Ref) | 63 (18.4) | 279 (81.6) | 1 | 1 | |
| | Service | 12 (44.4) | 15 (55.6) | 3.5 (1.5–7.9) | 3.5 (1.0–11.3) | 0.035* |
| **Socio economic status** | Lower class(Ref) | 17 (13.8) | 106 (86.2) | 1 | 1 | |
| | Middle class | 25 (19.4) | 104 (80.6) | 1.4 (0.7–2.9) | 1.5 (0.6–3.8) | 0.308 |
| | Upper class | 41 (30.4) | 94 (69.6) | 2.7 (1.4–5.1) | 1.1 (0.4–2.7) | 0.776 |
| **Distance to reach nearest health facility(n = 387)** | | | | 0.9 (0.9–0.9) | 0.9 (0.9–0.9) | 0.020* |
| **Place of delivery** | Home(Ref) | 6 (8.8) | 62 (91.2) | 1 | 1 | |
| | Health facility | 77 (24.1) | 242 (75.9) | 3.2(1.3–7.8) | 2.7 (0.8–9.2) | 0.092 |
| **Perceived severity** | Mild (Ref) | 5 (5.7) | 82 (94.3) | 1 | 1 | |
| | Moderate | 44(22.6) | 151(77.4) | 4.7 (1.8–12.5) | 7.6 (2.1–27.2) | 0.002* |
| | Severe | 34(32.4) | 71(67.6) | 7.8 (2.9–21.1) | 15.5 (3.4–69.3) | <0.001** |
| **History of difficulty breathing** | No (Ref) | 45 (18.1) | 203 (81.9) | 1 | 1 | |
| | Yes | 38 (27.3) | 101 (72.7) | 1.6 (1.0–2.7) | 1.7 (0.6–4.6) | 0.235 |
| **Knowledge on danger signs(n = 275)** | | | | 1.1(1.0–1.3) | 0.9 (0.8–1.1) | 0.903 |
| **Number of symptoms** | One (Ref) | 22 (22.7) | 75 (77.3) | 1 | 1 | |
| | Two | 13 (14.3) | 78 (85.7) | 0.5 (0.2–1.2) | 0.4 (0.1–1.2) | 0.11 |
| | ≥Three | 48 (24.1) | 151(75.9) | 1.0 (0.6–1.9) | 0.4 (0.1–1.2) | 0.1 |
| **Duration of illness** | ≤ 3 days (Ref) | 11 (12.8) | 75 (87.2) | 1 | 1 | |
| | 4 to 7 days | 29 (23.8) | 93 (76.2) | 2.1 (0.9–4.5) | 1.28 (0.4–3.5) | 0.626 |
| | >7days | 43 (24.0) | 136 (76.0) | 2.1 (1.0–4.4) | 0.4 (0.08–2.6) | 0.402 |
| **Number of children** | >2 children (Ref) | 72 (23.2) | 238 (76.8) | 1 | 1 | |
| | Up to 2 children | 11(14.3) | 66 (85.7) | 1.8 (0.9–3.6) | 1.6 (0.5–4.7) | 0.359 |

*p value <0.05,

**p value <0.001,

AOR = Adjusted Odds Ratio, COR = Crude Odds Ratio, Ref = Reference Category, n = number of samples.

## Discussion

This study revealed that only one quarter (25.3%) of the caretakers with ill under-five children sought health care from the health facility as the first source of care. This finding is consistent with the study findings from Rural Nigeria, Pokhara, North West Ethiopia [20–22]. This might be due to the pluralistic health system in the country where mothers or caretakers seek health care from various sources and do not visit the health facility until the illness become severe.

The current study showed that among the action taken during childhood illness the most common action (42.4%) was visiting pharmacy at first rather than going to a health facility. Different studies have shown that pharmacy is the most common source of health care seeking for childhood illness [5, 21, 23]. This might be due to the easy access to medicine form the pharmacy and caretakers do not have to pay a consultation fee for doctors. In the context of the Birendranagar municipality, there are abundant pharmacies as compared to the health

facility, pharmacies are present in each ward, so caretakers prefer to buy medicine directly from the pharmacy rather that going to the health facility.

In this study, the most common reason for not seeking any care for their children's illness was illness being mild and it would recover itself. This finding is consistent with the study findings from Yemen and North West Ethiopia, where reasons for not seeking medical care were illness being mild and illness would recover itself [22, 24].

This study has revealed that caretakers preferred private health facilities over government health facilities for treating the illness under-five children, Similar findings were shown by the studies from Nepal and Pakistan [5, 25]. Reasons for preferring private health facilities might be due to the availability of prompt care and caretaker's perception of good quality of health care at private health facilities.

The result of multivariate analysis showed that with the increase in distance from the health facilities caretakers were less likely to seek appropriate health care (AOR = 0.957, 95% CI, 0.923–0.993). This finding is consistent with the study findings from Ethiopia and a systematic review where caregivers located near to health care facilities were more likely to visit the health facility than those who lived far [26, 27]. The reason might be with the increase in distance from the health facility the transportation cost increases, it is difficult for caretakers to travel long distance with the child, so caretakers prefer to take the ill child to the nearby pharmacist or they try home remedies or they go to the nearby traditional healers. Most of the caretakers are mothers and they have many competing household duties to do at home so they do not take their child to a health facility unless illness becomes severe.

The current study revealed that caretakers who had gained secondary education were less likely to seek appropriate health care (AOR = 0.357, 95% CI 0.142–0.896) than those who were illiterate or had gained education up to the primary level. This might be due to the reason that caretakers with no education or primary education might become more conscious about the child's illness and they might not have knowledge about self- medication, so they took their child to health facility directly but among caretakers, with secondary education, they have little knowledge about medications, so they try self-medication for their child either by buying medicine from the pharmacy directly or might use different home remedies. This finding is similar to the study finding from rural India where parents with high school education and graduates were more likely to practice self -medication than illiterate parents [28]. Likewise, a study done in Kerela, India showed that mothers with higher educations were less likely to seek health care [29].

It was found that caretakers who perceived illness as severe were more likely to seek appropriate health care than those who perceived illness as mild. The possible reason might be when caretakers perceived their children have a severe illness they are more likely to seek care from health facilities to avoid further complications.

The current study showed a significant association between occupation and appropriate healthcare-seeking. Those caretakers who are involved in service as occupation were about four times more likely to seek appropriate health care (AOR = 3.533, 95% CI 1.096–11.384). This may be due to the reason that those caretakers who are involved in service are economically independent as well as have a higher education level. If their child gets ill they know that child should be taken immediately to the health facility and they can also afford to treat their child at the health facility.

## Limitation of the study

This study has some limitations; the answers provided by caretakers were based on the perception of caretakers about the illness of their children not based on exact medical diagnosis. Illnesses of only one month's duration were included to minimize the chances of recall bias.

## Conclusion

This study concluded that only one-quarter of the caretakers seek care from health facilities during their children's illness. Among healthcare-seeking behavior, the most common source was visiting the pharmacy directly at first rather than going to the health facility. Six out of ten health workers preferred private health facilities over government health facilities. Only four out of ten caretakers sought prompt health care. Caretakers had low awareness of danger signs of childhood illness. Four factors such as distance to reach the nearest health facility, education of the caretakers, perceived severity of the illness, and occupation of the caretakers were found significantly associated with healthcare-seeking behavior. Strong policy and regulations should be formulated and implemented at Birendranagar Municipality of Surkhet district to prevent direct purchase of medicines from pharmacies without any consultation. It is essential to conduct the health awareness program at community level on early recognition of danger signs and importance of consulting health facilities.

## Supporting information

**S1 Appendix. Data collection tool in English.**
(DOCX)

**S2 Appendix. Data collection tool in Nepali.**
(DOCX)

**S1 File. Data set underlying the findings of this study in SPSS.**
(SAV)

## Acknowledgments

We are indebted to all the Faculty members and staff of the Department of Community Medicine and Public Health, Institute of Medicine, Tribhuvan University for their continued encouragement and intellectual support at various stages in the completion of this study. We would like to express our deep gratitude to all caretakers of under-five children who participated in the study.

## Author Contributions

**Conceptualization:** Ganga Tiwari, Ganesh Tiwari, Durga Prasad Pahari.

**Formal analysis:** Ganga Tiwari.

**Investigation:** Ganga Tiwari, Ajoy Kumar Thakur.

**Methodology:** Ganga Tiwari, Ajoy Kumar Thakur, Durga Prasad Pahari.

**Resources:** Ganesh Tiwari.

**Software:** Sushil Pokhrel.

**Supervision:** Ajoy Kumar Thakur, Durga Prasad Pahari.

**Validation:** Ajoy Kumar Thakur, Sushil Pokhrel, Ganesh Tiwari, Durga Prasad Pahari.

**Visualization:** Ajoy Kumar Thakur, Sushil Pokhrel, Ganesh Tiwari, Durga Prasad Pahari.

**Writing – original draft:** Ganga Tiwari, Ganesh Tiwari.

**Writing – review & editing:** Ganga Tiwari, Ajoy Kumar Thakur, Sushil Pokhrel, Durga Prasad Pahari.

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
