## [Decision Letter · Decision Letter 0]

3 Sep 2020

PONE-D-20-18110

Factors associated with health care seeking behavior on perceived illness of under five year children among caretakers in Birendranagar municipality, Surkhet, Nepal

PLOS ONE

Dear Dr. Ganga Tiwari

Thank you for submitting your manuscript to PLOS ONE. After careful consideration, we feel that it has merit but does not fully meet PLOS ONE’s publication criteria as it currently stands. Therefore, we invite you to submit a revised version of the manuscript that addresses the points raised during the review process.

We look forward to receiving your revised manuscript.

Kind regards,

Edris Hasanpoor

Academic Editor

PLOS ONE

Additional Editor Comments:

1.Table 6, you should list the number of samples for each variable category.

2.Table 6, the definition of Ethnicity is not explained.

3.Table 6, is the secondary education includes higher secondary within the education level?

4.Why does Table 6 list the service industry separately? The service industry only accounts for 7%.

5.Table 6, i don't see the definition of socioeconomic status classification.

6.Does the distance between Table 6 and health care facilities refer to the geographical distance or the travel time as in Table 2?

7.Table 6, about the history of difficult breathing, the definition of difficulty?

8.What is the operational definition of knowledge for dangerous signals in Table 6?

9.From Table 2 we know that the reasons for choosing public and private medical institutions are different, and the regression analysis of seeking medical behaviors can be considered as a stratified analysis.

10.It is recommended that the coding method of the dependent variable be presented below Table 6 for explanation.

11. Is it confirmed that there are no multicollinearity issues.

12.The quality of logistic regression model and the result about Hosmer-Lemeshow Test?(less...)

Generally, it is a useful paper since it addresses one of the major problems in developing countries, the relatively high death-rate of new-borns and small children. Unfortunately, however, there are some shortcomings which should be addressed before accepting this paper for publication.

The authors refer to several studies in the Discussion chapter, sometimes to one or two, other times to other one or two, etc, but no general overview of these studies has been given. Probably the whole paper could be made shorter if a summary (maybe even in a table format) would be presented in the Introduction, and the major points would be stated. Later, then, they only have tp refer to this section.

Methods are poorly described, especially regarding the interviews. No detailed information is given about the questions of the interviews. At least some typical examples and variations should be presented (maybe as an Appendix). The interview method is typically a crucial information since it can manipulate the interviewees and can lead to false interpretation. It isn’t clear, too, what method for coding had been used.

It seems unusual to use 0.2 p-value for statistical analysis; at least some explanation should be given.

Discussion is too long, generally because results are repeated several times in this section. Generally, Discussion is an overview of the results obtained is a short and compact format, to give the reader an impression about the meaning of the research. The many references to other studies make it even more complicated (see above).

A technical note: No explanation is given on the first mention for abbreviation ARI (Page 4), that of FCHV (Page 5), IMCI (Page 6). etc. Although there is a list of abbreviations at the end of the manuscript, the reader does not find it while reading the text. Either the table should be placed onto the beginning of the paper, or abbreviations should be explained when first mentioned.

Overall opinion: it is an important topic, but the manuscript requires general improvement (major revision) before accepting.

Journal Requirements:

5. Please ensure that you refer to Figure 1 in your text as, if accepted, production will need this reference to link the reader to the figure.

6. We note you have included a table to which you do not refer in the text of your manuscript. Please ensure that you refer to Table 6 in your text; if accepted, production will need this reference to link the reader to the Table.

Reviewers' comments:

Reviewer's Responses to Questions

**Comments to the Author**

1. Is the manuscript technically sound, and do the data support the conclusions?

Reviewer #1: Partly

Reviewer #2: Partly

2. Has the statistical analysis been performed appropriately and rigorously? 

Reviewer #1: Yes

Reviewer #2: No

3. Have the authors made all data underlying the findings in their manuscript fully available?

Reviewer #1: No

Reviewer #2: No

4. Is the manuscript presented in an intelligible fashion and written in standard English?

Reviewer #1: No

Reviewer #2: Yes

5. Review Comments to the Author

Reviewer #1: Generally, it is a useful paper since it addresses one of the major problems in developing countries, the relatively high death-rate of new-borns and small children. Unfortunately, however, there are some shortcomings which should be addressed before accepting this paper for publication.

The authors refer to several studies in the Discussion chapter, sometimes to one or two, other times to other one or two, etc, but no general overview of these studies has been given. Probably the whole paper could be made shorter if a summary (maybe even in a table format) would be presented in the Introduction, and the major points would be stated. Later, then, they only have tp refer to this section.

Methods are poorly described, especially regarding the interviews. No detailed information is given about the questions of the interviews. At least some typical examples and variations should be presented (maybe as an Appendix). The interview method is typically a crucial information since it can manipulate the interviewees and can lead to false interpretation. It isn’t clear, too, what method for coding had been used.

It seems unusual to use 0.2 p-value for statistical analysis; at least some explanation should be given.

Discussion is too long, generally because results are repeated several times in this section. Generally, Discussion is an overview of the results obtained is a short and compact format, to give the reader an impression about the meaning of the research. The many references to other studies make it even more complicated (see above).

A technical note: No explanation is given on the first mention for abbreviation ARI (Page 4), that of FCHV (Page 5), IMCI (Page 6). etc. Although there is a list of abbreviations at the end of the manuscript, the reader does not find it while reading the text. Either the table should be placed onto the beginning of the paper, or abbreviations should be explained when first mentioned.

Overall opinion: it is an important topic, but the manuscript requires general improvement (major revision) before accepting.

Reviewer #2: 1.Table 6, you should list the number of samples for each variable category.

2.Table 6, the definition of Ethnicity is not explained.

3.Table 6, is the secondary education includes higher secondary within the education level?

4.Why does Table 6 list the service industry separately? The service industry only accounts for 7%.

5.Table 6, i don't see the definition of socioeconomic status classification.

6.Does the distance between Table 6 and health care facilities refer to the geographical distance or the travel time as in Table 2?

7.Table 6, about the history of difficult breathing, the definition of difficulty?

8.What is the operational definition of knowledge for dangerous signals in Table 6?

9.From Table 2 we know that the reasons for choosing public and private medical institutions are different, and the regression analysis of seeking medical behaviors can be considered as a stratified analysis.

10.It is recommended that the coding method of the dependent variable be presented below Table 6 for explanation.

11. Is it confirmed that there are no multicollinearity issues.

12.The quality of logistic regression model and the result about Hosmer-Lemeshow Test?

6. PLOS authors have the option to publish the peer review history of their article (what does this mean?). If published, this will include your full peer review and any attached files.

Reviewer #1: No

Reviewer #2: No

---

## [Author Response · Author response to Decision Letter 0]

5 Oct 2020

Response to the reviewer and editor

Date: October 5, 2020

To the Editor, 

PLOS ONE Journal

Greetings! Hope you are fine and doing well. First of all, I would like to thank both the editor and reviewers for the extensive review of our manuscript and providing us opportunities to revise it. I have tried to incorporate all the feedback provided to us. For the revision of this manuscript, I took the support of my colleagues Dr. Sushil Pokhrel and Dr. Ganesh Tiwari. I checked language, spellings, and grammar using software/app for made for checking grammar. I have manually checked each sentence of this manuscript. I have mentioned below a table that explains my response in each comment of the reviewers and editor. Now, I am submitting the revised version of the manuscript as well as the original version with track changes. I am also submitting the supporting information such as tools of data collection in English as well as Nepali language and SPSS data set. Thank you very much for your time and consideration.

Best regards!

Ganga Tiwari

SN. Comments by editor and reviewer Response

 Editor Comments 

1 Table 6, you should list the number of samples for each variable category 

- Yes, I have listed the number of samples for each variable

2 Table 6, the definition of Ethnicity is not explained 

- I have provided an operational definition of ethnicity. 

3 Table 6, is that secondary education includes higher secondary within the education level?

 - No the secondary education includes lower secondary education (6th standard to 10th standard) in our study.

Higher secondary is considered a college education in our study.

4 Why does Table 6 list the service industry separately? The service industry only accounts for 7%. 

- I have checked the association of healthcare-seeking behavior with all the occupations categorizing into two group like home makers and non-homemakers/ business and non-business/agriculture and non-agriculture and service and non-service, I did not found any association among other categories but I found association with service and non -service category, so I have listed service and non-service category in table 6.

5 In table 6, I don't see the definition of socioeconomic status classification. 

- Now, I have mentioned the definition of socioeconomic class in the operational definition.

6 Does the distance between Table 6 and health care facilities refer to the geographical distance or the travel time as in Table 2?

-Yes, the distance mentioned in table 6 refers to the geographical distance which is measured in travel time as in table 2. But in Bivariate and multivariate analysis I have not categorized distance into below 30 min and above 30 min, as the distance is a continuous variable I directly check the association of distance and healthcare-seeking behavior. It was measured in travel time to reach the nearest health facility in minutes. 

7. Table 6, about the history of difficult breathing, the definition of difficulty? 

- Now, I have mentioned about the definition of difficulty in breathing in the operational definition. 

8. What is the operational definition of knowledge for dangerous signals in Table 6?

-Now, I have mentioned the definition of knowledge of danger signs in table 6 at the operational definition section.

9 From, Table 2 we know that the reasons for choosing public and private medical institutions are different, and the regression analysis of seeking medical behaviors can be considered as homemakers analysis. Yes, the regression analysis of seeking medical behavior can be considered as a stratified analysis.

10 It is recommended that the coding method of the dependent variable be presented below Table 6 for an explanation. 

- Now, I have mentioned in the data analysis section under methodology.

11 Is it confirmed that there are no multicollinearity issues? 

- Yes, I have calculated the Variance Inflation Factor (VIF) among the independent variables selected in the multivariate logistic analysis model. VIF value was less than 2 for every variable checked. So it is confirmed that there is no issue of multicollinearity.

12 The quality of the logistic regression model and the result of the Hosmer-Lemeshow Test? (less...)

-Hosmer Lemeshow test, the goodness of fit of the model was assessed. The test statistic was 0.500 (>0.05) that showed that the model adequately fits the data.

 The authors refer to several studies in the Discussion chapter, sometimes to one or two, other times to other one or two, etc, but no general overview of these studies has been given. Probably the whole paper could be made shorter if a summary (maybe even in a table format) would be presented in the Introduction, and the major points would be stated. Later, then, they only have to refer to this section.

 - Now, I have tried to make the discussion section shorter by reducing the repeated contents of result and mentioned only major issues. 

 Methods are poorly described, especially regarding the interviews. No detailed information is given about the questions of the interviews. At least some typical examples and variations should be presented (maybe as an Appendix). The interview method is typically a crucial information since it can manipulate the interviewees and can lead to false interpretation. It isn’t clear, too, what method for coding had been used.

 - Now, I have explained the interview technique in the methodology part. Regarding the coding of the dependent variable, I have now explained in the operational definition.

 A technical note: No explanation is given on the first mention for abbreviation ARI (Page 4), that of FCHV (Page 5), IMCI (Page 6) etc. Although there is a list of abbreviations at the end of the manuscript, the reader does not find it while reading the text. Either the table should be placed onto the beginning of the paper, or abbreviations should be explained when first mentioned.

Overall opinion: It is an important topic, but the manuscript requires general improvement (major revision) before accepting. 

- Now I have explained abbreviations when used at first and then I have kept list of abbreviations at the last section of manuscript.

 Journal requirement 

 Please ensure that your manuscript meets PLOS ONE's style requirements, including those for file naming 

 - Now I have revised my manuscript using PLOS ONE’s style requirement, and I have named file name accordingly.

We suggest you thoroughly copyedit your manuscript for language usage, spelling, and grammar. If you do not know anyone who can help you do this, you may wish to consider employing a professional scientific editing service.

- Yes, I have thoroughly copyedited my manuscript for language, spelling, and grammar through grammar checking websites/apps as well as with the help of my colleagues.

 It seems unusual to use 0.2 p-values for statistical analysis; at least some explanation should be given.

Discussion is too long, generally because results are repeated several times in this section. Generally, Discussion is an overview of the results obtained is a short and compact format, to give the reader an impression about the meaning of the research. The many references to other studies make it even more complicated (see above). 

- It has been shown in a study that use of p-value < 0.05 is too stringent and often excludes important variables from the logistic regression model, so choosing a p-value ranging from 0.15 to 0.20 is highly recommended, various published studies have used p-value 0.2 for selecting the variables for multivariate logistic regression. Now I have explained this in the manuscript also.

Now I have tried to make the discussion section shorter by reducing the repeated contents of result and mentioned only major issues

---

## [Decision Letter · Decision Letter 1]

2 Feb 2021

PONE-D-20-18110R1

Factors associated with health care seeking behavior on the perceived illness of under five year children among caretakers in Birendranagar municipality, Surkhet, Nepal

PLOS ONE

Dear Dr. Tiwari,

Thank you for submitting your manuscript to PLOS ONE. After careful consideration, we feel that it has merit but does not fully meet PLOS ONE’s publication criteria as it currently stands. Therefore, we invite you to submit a revised version of the manuscript that addresses the points raised during the review process.

Thank you for addressing all the comments on the previous version. The manuscript is is a good shape at the moment. We only have some minor comments to further improve the language and readability of the manuscript. Hope to receive the revised manuscript soon.

We look forward to receiving your revised manuscript.

Kind regards,

Shyam Sundar Budhathoki, MBBS, MD, MPH

Academic Editor

PLOS ONE

Reviewers' comments:

Reviewer's Responses to Questions

6. Review Comments to the Author

Reviewer #3: Title: needs to be modified, can be made short

Abstract: Objective needs to be more specific. Conclusion needs language editing, can be made short. Last statement not a part of study objective.

Keywords: can be written as per MeSH terms

Introduction: needs language editing ; can be made short.

Objectives: need to be more specific

Methods: section needs to be rewritten. It is very lengthy, can be made short with only relevant details. Sample size and technique can be explained by a flowchart; will make it concise and reflect better understanding. “FCHVs” needs to be spelt out for the first time in text, it can later be used as an abbreviation. Was Kuppuswami scale modified Nepali version validated, before using in study. “Perceived Illness” used in study can be elaborated in operational definition section. Inclusion criteria mentions about only the treatment seeking period of last one month, which is a major limitation. Childhood/under-five manifestations may not always be present, a longer duration would be better like 3 months. Sentence construction not proper, grammatical errors need to be corrected. Pretesting of study tool mentioned, but whether the same sample was included in the sample size or not, is not mentioned.

Results: require language editing.

“Perceived Illness” needs to be explained in methodology. Similarly, “Perceived Severity” mentioned not explained in text. Too many[Table 1 to 5] tables. Can be combined with statistical analysis part of bivariable analysis. Multivariable analysis[Table 6] is fine.

Discussion: is very elaborate; can be shortened in length and only relevant findings highlighted.

Limitations: not mentioned

References: Some old references may be removed. Recent studies within last 5 years can be cited.

---

## [Author Response · Author response to Decision Letter 1]

22 Mar 2021

Reviewer #3: Title: needs to be modified, can be made short

-The title has been modified and made short

Abstract: Objective needs to be more specific. Conclusion needs language editing, can be made short. Last statement not a part of study objective 

-Now, objectives are written more specifically, conclusion section has been edited.

Keywords: can be written as per MeSH terms 

-Now, Key words are written as per MeSH terms

Objectives: need to be more specific 

-Now, the objectives are written more specifically.

Methods: section needs to be rewritten. It is very lengthy, can be made short with only relevant details. Sample size and technique can be explained by a flowchart; will make it concise and reflect better understanding. “FCHVs” needs to be spelt out for the first time in text, it can later be used as an abbreviation. Was Kuppuswami scale modified Nepali version validated, before using in study. “Perceived Illness” used in study can be elaborated in operational definition section. Inclusion criteria mentions about only the treatment seeking period of last one month, which is a major limitation. Childhood/under-five manifestations may not always be present, a longer duration would be better like 3 months 

-We have rewritten methodology section. We have tried to make it short. Sampling technique is now explained by a flowchart. FCHVs are now written in full word when it is first used.

Kuppuswamy scale modified in context of Nepal was was used; this version was already validated in context of Nepal and used by various published studies.

Perceived illness is now defined on the operational definition section.

Illness period of only one month duration was taken to avoid recall bias among the caretakers.

Results: require language editing.

“Perceived Illness” needs to be explained in methodology. Similarly, “Perceived Severity” mentioned not explained in text. Too many[Table 1 to 5] tables. Can be combined with statistical analysis part of bivariable analysis. Multivariable analysis[Table 6] is fine 

-Perceived illness and severity has been defined in the operational definition section. Some tables have been removed, language editing has been done in result section.

Discussion: is very elaborate; can be shortened in length and only relevant findings highlighted ---- 

-Discussion section has been made short.

Limitations: not mentioned

-Now, limitation has been mentioned.

References: Some old references may be removed. Recent studies within last 5 years can be cited. 

-Some old references have been removed.

---

## [Decision Letter · Decision Letter 2]

9 Jun 2021

PONE-D-20-18110R2

Health care seeking behavior for common childhood illnesses in Birendranagar municipality, Surkhet, Nepal

PLOS ONE

Dear Dr. Tiwari,

Thank you for submitting your manuscript to PLOS ONE. After careful consideration, we feel that it has merit but does not fully meet PLOS ONE’s publication criteria as it currently stands. Therefore, we invite you to submit a revised version of the manuscript that addresses the points raised during the review process.

We look forward to receiving your revised manuscript.

Kind regards,

Shyam Sundar Budhathoki, MBBS, MD, MPH

Academic Editor

PLOS ONE

Journal Requirements:

Additional Editor Comments (if provided):

Reviewers' comments:

Reviewer's Responses to Questions

**Comments to the Author**

1. If the authors have adequately addressed your comments raised in a previous round of review and you feel that this manuscript is now acceptable for publication, you may indicate that here to bypass the “Comments to the Author” section, enter your conflict of interest statement in the “Confidential to Editor” section, and submit your "Accept" recommendation.

Reviewer #4: (No Response)

2. Is the manuscript technically sound, and do the data support the conclusions?

Reviewer #4: Yes

3. Has the statistical analysis been performed appropriately and rigorously? 

Reviewer #4: Yes

4. Have the authors made all data underlying the findings in their manuscript fully available?

Reviewer #4: Yes

5. Is the manuscript presented in an intelligible fashion and written in standard English?

Reviewer #4: Yes

6. Review Comments to the Author

Reviewer #4: Health care seeking behavior for common childhood illnesses in Birendranagar municipality, Surkhet, Nepal

Reviewer: Wingston Ng’ambi (Lecturer and Research Scientist), College of Medicine- Kamuzu University of Health Sciences (Formerly University of Malawi), Lilongwe, Malawi; & Institute of Global Health, University of Geneva, Geneva, Switzerland

This is a nice paper touching on an important area.

1. The title should be changed to “Factors associated with healthcare seeking behavior for common illnesses amongst under-five children in Surkhet district, Nepal: 2018“

2. The manuscript needs to be numbered in order to reference to line numbers. This makes review easier.

Abstract

Introduction of abstract

3. “There are few studies on health care seeking behavior among caretakers in Nepal, so the objective of this study was to itsassess the healthcare-seeking behavior of the caretakers in Birendranagar municipality of Surkhet district. So, this study aims to identify prevailing health care seeking behavior of caretakers on perceived illness of under five year children and to identify the association of socio demographic, economic, illness related and health system related factors with health care seeking behavior.” Should be changed to “There are few studies on healthcare seeking behavior among caretakers in Nepal. Therefore, we conducted this study to determine the level of healthcare seeking behavior of caretakers on perceived illness and to identify the factors associated with health care seeking behavior of the caretakers of under five-year children from Surkhet district in Nepal in 2018.”

Methods section of abstract

4. Add level of P-value for statistical significance

5. How the logistic regression models were built (optional depending on word count)

Results

6. Change “Regarding healthcare-seeking behavior of caretakers for their children’s illnesses, the most common source of care-seeking was visiting pharmacy directly at first (42.4 %), only one quarter (25.3 %) of the caretakers visited health facilities, among those who visited health facilities, only (37.2 %) of caretakers sought prompt health care.” to “Of these, 42.4% visited the pharmacy directly, 25.3% visited the health facilities and XXXX did nothing. Amongst those who visited a health facility, 37.2% of caretakers sought prompt health care.”

Conclusion

7. Rather than just repeat the results consider using statements like (“There is a need to understand and address individual and socio-economic barriers to health seeking to increase access and use of health care and fast-track progress towards Universal Health Coverage amongst children from Surkhet district in Nepal.” This is what would be an appropriate conclusion that answers the question: How does this study relate to the SDGs and universal health coverage as well as Nepal National Health Strategy?

Main body of the paper

8. Include a running title of the paper

Introduction

9. Change the objectives as stipulated in the abstract

Methods

10. p-value should be changed to P-value

11. The authors should make it clear how they arrived at the final model. Did they use log likelihood ratio methods or some other methods? They also need to be clear on what multivariate analysis that they are conducting.

12. Under a subsection “Appropriate health care seeking behavior” several studies include obtaining care from pharmacies as appropriate health care seeking behaviour (HSB), how come you are doing this differently?

13. Can you refer to “Ng'ambi W, Mangal T, Phillips A, Colbourn T, Mfutso-Bengo J, Revill P, Hallett TB. Factors associated with healthcare seeking behaviour for children in Malawi: 2016. Trop Med Int Health. 2020 Dec;25(12):1486-1495. doi: 10.1111/tmi.13499. Epub 2020 Oct 19. PMID: 32981174.” for some operational definition of HSB.

Results

14. Format the tables for characteristics to look like this.

Patient characteristics n (%)

Total 255229 (100.0)

Gender

Male 122610 (48.0)

Female 125275 (49.1)

Missing 7344 (2.9)

Location

Rural 168258 (65.9)

Urban 86971 (34.1)

Age at Sample draw (in months)

0-1 145622 (57.1)

2-5 74707 (29.3)

6-11 21307 (8.4)

12-17 3337 (1.3)

18-24 1902 (0.8)

Missing 8354 (3.3)

Region

Northern 22897 (9.0)

Central 72,633 (28.5)

Southern 159699 (62.6)

Year Sample drawn

2013 16308 (6.4)

2014 25858 (10.1)

2015 41271 (16.2)

2016 41178 (16.1)

2017 42252 (16.6)

2018 43370 (17.0)

2019 36372 (14.3)

2020* 7741 (3.0)

Missing 879 (0.3)

15. In all tables do not combine the mean or SD if the headings are number and percentage

16. All tables should not have lines crossing them

17. Include the illnesses (like fever, diarrheas) as appriori variables in the final multivariate model as these form key part of your analysis. Also include age and sex of the child as appriori variables.

18. We are not sure of the P-value for Table 5. Is it for crude or adjusted estimates? I would format the table to look like:-

Characteristics (n=26386) Bivariate analysis Multivariate analysis

OR (95%CI) P-value OR (95%CI) P-value

Age group

15-19 1.00 1.00

20-24 1.10 (1.00-1.21) 0.06 1.26 (1.13-1.41) <0.001

25-29 1.07 (0.96-1.18) 0.22 1.42 (1.24-1.62) <0.001

30-34 1.06 (0.95-1.18) 0.33 1.48 (1.27-1.72) <0.001

35-39 1.07 (0.95-1.21) 0.27 1.61 (1.36-1.92) <0.001

40-44 0.91 (0.78-1.07) 0.25 1.47 (1.19-1.81) <0.001

45-49 1.02 (0.80-1.30) 0.88 1.77 (1.33-2.34) <0.001

Year

2004/5 1.00 1.00

2010 1.23 (1.14-1.32) <0.001 1.21 (1.12-1.31) <0.001

2015/16 2.19 (2.03-2.37) <0.001 2.12 (1.97-2.29) <0.001

Region

North 1.00

Centre 1.03 (0.94-1.12) 0.55

South 0.96 (0.88-1.05) 0.40

Number of previous children ever born

1 1.00 1.00

2-3 0.79 (0.74-0.85) <0.001 0.70 (0.64-0.76) <0.001

4-5 0.74 (0.68-0.80) <0.001 0.62 (0.55-0.70) <0.001

6+ 0.69 (0.63-0.75) <0.001 0.59 (0.51-0.69) <0.001

Education level

None 1.00 1.00

Primary 1.22 (1.13-1.32) <0.001 1.09 (1.00-1.18) 0.05

Secondary 1.69 (1.54-1.86) <0.000 1.24 (1.11-1.39) <0.001

Tertiary 4.36 (3.42-5.57) <0.001 2.35 (1.80-3.06) <0.001

Wealth index quintile

Poorest 1.00 1.00

Poorer 1.07 (0.99-1.16) 0.10 1.07 (0.99-1.17) 0.09

Middle 1.09 (1.00-1.18) 0.05 1.10 (1.01-1.20) 0.024

Richer 1.17 (1.07-1.28) <0.001 1.15 (1.05-1.26) 0.002

Richest 1.47 (1.35-1.60) <0.001 1.23 (1.11-1.36) <0.001

Residence

Urban 1.00

Rural 0.78 (0.72-0.84) <0.001

Sources of antenatal care knowledge

Frequency of listening to radio

Less than once a week 1.00

At least once a week 1.00 (0.95-1.05) 0.97

Frequency of watching television

Less than once a week 1.00 1.00

At least once a week 1.44 (1.31-1.58) <0.001 1.13 (1.02-1.26) 0.025

Barriers to access antenatal care

Permission to visit health services

No problem 1.00

Big problem 1.10 (1.02-1.19) 0.01

Money to pay for health services

No problem 1.00

Big problem 0.88 (0.83-0.92) <0.001

Distance to health facilities

No problem 1.00

Big problem 0.93 (0.88-0.98) 0.005

Presence of companion

No problem 1.00

Big problem 0.98 (0.92-1.04) 0.50

No drugs at health facility

No problem 1.00

Big problem 1.28 (1.21-1.35) <0.001

No female provider

No problem 1.00

Big problem 1.01 (0.95-1.09) 0.64

Marital status

Never married 1.00 1.00

Married 1.20 (1.02-1.41) 0.026 1.65 (1.39-1.96) <0.001

Widowed 1.01 (0.78-1.34) 0.89 1.50 (1.13-1.99) 0.005

Divorced 1.08 (0.90-1.29) 0.40 1.50 (1.24-1.81) <0.001

Discussion

19. Start the discussion with a summary of the key results of your study and then move on to discuss them in light of other relevant literature.

20. The findings from this paper may be helpful to your discussion and conclusion. Consider citing it as well.

Ng'ambi W, Mangal T, Phillips A, Colbourn T, Mfutso-Bengo J, Revill P, Hallett TB. Factors associated with healthcare seeking behaviour for children in Malawi: 2016. Trop Med Int Health. 2020 Dec;25(12):1486-1495. doi: 10.1111/tmi.13499. Epub 2020 Oct 19. PMID: 32981174.

21. Don’t repeat results in the discussion but rather compare your key results with those of other studies and explain any disparity or agreement.

Conclusion

22. This section should provide policy insights as well as alignment of the study findings with the SDGs as well as universal health coverage.

23. Include the policy relevance of your study findings as well.

7. PLOS authors have the option to publish the peer review history of their article (what does this mean?). If published, this will include your full peer review and any attached files.

Reviewer #4: **Yes: **Wingston Felix Ng’ambi

---

## [Author Response · Author response to Decision Letter 2]

31 Jul 2021

Response to the reviewer and editor

Date: July 31, 2020

To the Editor, 

PLOS ONE Journal

Greetings! Hope you are fine and doing well. First of all, I would like to thank both the editor and reviewers for the extensive review of our manuscript and providing us opportunities to revise it. For the revision of this manuscript, I took the support of my colleagues Dr. Sushil Pokhrel and Dr. Ganesh Tiwari. We have tried to incorporate all the feedback provided in the revised version of the manuscript. I have mentioned my response in comment of the reviewers and editor. Now, I am submitting the revised version of the manuscript as well as the original version with track changes. Thank you very much for your time and consideration.

Best regards!

Ganga Tiwari

Journal Requirements:

Author’s response: I have reviewed the reference list. It is complete and do not include the retracted article,

Reviewer's Responses to Questions

1. The title should be changed to “Factors associated with healthcare seeking behavior for common illnesses amongst under-five children in Surkhet district, Nepal: 2018“

 Author’s response: I have changed the title accordingly.

2. The manuscript needs to be numbered in order to reference to line numbers. This makes review easier.

Abstract

Introduction of abstract

3. “There are few studies on health care seeking behavior among caretakers in Nepal, so the objective of this study was to itsassess the healthcare-seeking behavior of the caretakers in Birendranagar municipality of Surkhet district. So, this study aims to identify prevailing health care seeking behavior of caretakers on perceived illness of under five year children and to identify the association of socio demographic, economic, illness related and health system related factors with health care seeking behavior.” Should be changed to “There are few studies on healthcare seeking behavior among caretakers in Nepal. Therefore, we conducted this study to determine the level of healthcare seeking behavior of caretakers on perceived illness and to identify the factors associated with health care seeking behavior of the caretakers of under five-year children from Surkhet district in Nepal in 2018.”

Author’s response: I have made slight change in the language of the objective and changed the objective, we did not determine the level, and we only determine the different health care seeking behaviors among the care takers.

Methods section of abstract

4. Add level of P-value for statistical significance

Author’s response: Previously, I had added level of p- value for statistical significance in abstract but previous reviewers of this journal suggested me to remove it from abstract so I have removed it. 

5. How the logistic regression models were built (optional depending on word count)

Author’s response: Abstract become lengthy, if we add this detail

Results

6. Change “Regarding healthcare-seeking behavior of caretakers for their children’s illnesses, the most common source of care-seeking was visiting pharmacy directly at first (42.4 %), only one quarter (25.3 %) of the caretakers visited health facilities, among those who visited health facilities, only (37.2 %) of caretakers sought prompt health care.” to “Of these, 42.4% visited the pharmacy directly, 25.3% visited the health facilities and XXXX did nothing. Amongst those who visited a health facility, 37.2% of caretakers sought prompt health care.”

Author’s response: Thank you for your feedback; I have changed the result section accordingly.

Conclusion

7. Rather than just repeat the results consider using statements like (“There is a need to understand and address individual and socio-economic barriers to health seeking to increase access and use of health care and fast-track progress towards Universal Health Coverage amongst children from Surkhet district in Nepal.” This is what would be an appropriate conclusion that answers the question: How does this study relate to the SDGs and universal health coverage as well as Nepal National Health Strategy?

Author’s response: Since, this study is confined to Birendranagar Municipality of Surkhet district, we have concluded according to our study results at the municipality level. Now, we have also included policy implication of our study.

Main body of the paper

8. Include a running title of the paper

Introduction

9. Change the objectives as stipulated in the abstract

Methods

10. p-value should be changed to P-value

11. The authors should make it clear how they arrived at the final model. Did they use log likelihood ratio methods or some other methods? They also need to be clear on what multivariate analysis that they are conducting.

 Author’s response: 

We used SPSS for the data analysis. The twelve variables (ethnicity, educational status, occupation, socioeconomic status, distance to reach nearest health facility, place of delivery, perceived severity, history of difficulty breathing, knowledge on danger signs, number of symptoms, duration of illness, number of children) that exhibited significant association with health care seeking in bivariate analysis i.e. p value ≤ 0.2 were further analyzed in multivariate analysis using binomial logistic regression. Multivariate analysis was done for adjustment of possible confounders. Hosmer Lemeshow test, the goodness of fit of the model was assessed. The test statistic was 0.500 (>0.05) that showed that the model adequately fits the data.

12. Under a subsection “Appropriate health care seeking behavior” several studies include obtaining care from pharmacies as appropriate health care seeking behaviour (HSB), how come you are doing this differently?

 Author’s response: We also took references of several studies, obtaining medicines from pharmacies without prescription/doctor’s consultation is considered as in-appropriate health care seeking behavior. Obtaining medicines from pharmacies without any consultation comes under self- medication, because of this irrational use of medication; antibiotic/drug resistance is a big issue in Nepal, We had taken the reference of studies from Yemen, India and North- West Ethiopia. Full citations of these studies are mentioned below.

Kalita D, Borah M, Kakati R, Borah H. Primary Caregivers Health Seeking Behaviour for Under-Five Children : A Study in a Rural Block of Assam, India. Ntl J Community Med. 2016;7(11):868–72. 

 Webair HH, Bin-Gouth AS. Factors affecting health seeking behavior for common childhood illnesses in Yemen. Patient Prefer Adherence. 2013;7:1129–38. 

Molla Simieneh M, Mengistu Y, Gelagay AA, Gebeyehu MT. Mothers’ health care seeking behavior and associated factors for common childhood illnesses, Northwest Ethiopia: community based cross-sectional study. 

13. Can you refer to “Ng'ambi W, Mangal T, Phillips A, Colbourn T, Mfutso-Bengo J, Revill P, Hallett TB. Factors associated with healthcare seeking behaviour for children in Malawi: 2016. Trop Med Int Health. 2020 Dec;25(12):1486-1495. doi: 10.1111/tmi.13499. Epub 2020 Oct 19. PMID: 32981174.” for some operational definition of HSB.

Author’s response: As we had conducted this study on 2018, we had defined operational definitions ourselves before the conduction of study taking reference of relevant literature, now study have already conducted, we can’t modify our operational definitions. Operational definitions are always defined before the conduction of study, not after the completion of study, as operational definitions involves variables; these are set before data collection.

Results

14. Format the tables for characteristics to look like this.

Patient characteristics n (%)

Total 255229 (100.0)

Gender

Male 122610 (48.0)

Female 125275 (49.1)

Missing 7344 (2.9)

Location

Rural 168258 (65.9)

Urban 86971 (34.1)

Age at Sample draw (in months)

0-1 145622 (57.1)

2-5 74707 (29.3)

6-11 21307 (8.4)

12-17 3337 (1.3)

18-24 1902 (0.8)

Missing 8354 (3.3)

Region

Northern 22897 (9.0)

Central 72,633 (28.5)

Southern 159699 (62.6)

Year Sample drawn

2013 16308 (6.4)

2014 25858 (10.1)

2015 41271 (16.2)

2016 41178 (16.1)

2017 42252 (16.6)

2018 43370 (17.0)

2019 36372 (14.3)

2020* 7741 (3.0)

Missing 879 (0.3)

15. In all tables do not combine the mean or SD if the headings are number and percentage

Author’s response: Thank you for your feedback, I have updated the tables accordingly.

16. All tables should not have lines crossing them

Author’s response: Thank you for your feedback, I have updated the table accordingly.

17. Include the illnesses (like fever, diarrheas) as appriori variables in the final multivariate model as these form key part of your analysis. Also include age and sex of the child as appriori variables.

18. We are not sure of the P-value for Table 5. Is it for crude or adjusted estimates? I would format the table to look like:-

Characteristics (n=26386) Bivariate analysis Multivariate analysis

OR (95%CI) P-value OR (95%CI) P-value

Age group

15-19 1.00 1.00

20-24 1.10 (1.00-1.21) 0.06 1.26 (1.13-1.41) <0.001

25-29 1.07 (0.96-1.18) 0.22 1.42 (1.24-1.62) <0.001

30-34 1.06 (0.95-1.18) 0.33 1.48 (1.27-1.72) <0.001

35-39 1.07 (0.95-1.21) 0.27 1.61 (1.36-1.92) <0.001

40-44 0.91 (0.78-1.07) 0.25 1.47 (1.19-1.81) <0.001

45-49 1.02 (0.80-1.30) 0.88 1.77 (1.33-2.34) <0.001

Year

2004/5 1.00 1.00

2010 1.23 (1.14-1.32) <0.001 1.21 (1.12-1.31) <0.001

2015/16 2.19 (2.03-2.37) <0.001 2.12 (1.97-2.29) <0.001

Region

North 1.00

Centre 1.03 (0.94-1.12) 0.55

South 0.96 (0.88-1.05) 0.40

Number of previous children ever born

1 1.00 1.00

2-3 0.79 (0.74-0.85) <0.001 0.70 (0.64-0.76) <0.001

4-5 0.74 (0.68-0.80) <0.001 0.62 (0.55-0.70) <0.001

6+ 0.69 (0.63-0.75) <0.001 0.59 (0.51-0.69) <0.001

Education level

None 1.00 1.00

Primary 1.22 (1.13-1.32) <0.001 1.09 (1.00-1.18) 0.05

Secondary 1.69 (1.54-1.86) <0.000 1.24 (1.11-1.39) <0.001

Tertiary 4.36 (3.42-5.57) <0.001 2.35 (1.80-3.06) <0.001

Wealth index quintile

Poorest 1.00 1.00

Poorer 1.07 (0.99-1.16) 0.10 1.07 (0.99-1.17) 0.09

Middle 1.09 (1.00-1.18) 0.05 1.10 (1.01-1.20) 0.024

Richer 1.17 (1.07-1.28) <0.001 1.15 (1.05-1.26) 0.002

Richest 1.47 (1.35-1.60) <0.001 1.23 (1.11-1.36) <0.001

Residence

Urban 1.00

Rural 0.78 (0.72-0.84) <0.001

Sources of antenatal care knowledge

Frequency of listening to radio

Less than once a week 1.00

At least once a week 1.00 (0.95-1.05) 0.97

Frequency of watching television

Less than once a week 1.00 1.00

At least once a week 1.44 (1.31-1.58) <0.001 1.13 (1.02-1.26) 0.025

Barriers to access antenatal care

Permission to visit health services

No problem 1.00

Big problem 1.10 (1.02-1.19) 0.01

Money to pay for health services

No problem 1.00

Big problem 0.88 (0.83-0.92) <0.001

Distance to health facilities

No problem 1.00

Big problem 0.93 (0.88-0.98) 0.005

Presence of companion

No problem 1.00

Big problem 0.98 (0.92-1.04) 0.50

No drugs at health facility

No problem 1.00

Big problem 1.28 (1.21-1.35) <0.001

No female provider

No problem 1.00

Big problem 1.01 (0.95-1.09) 0.64

Marital status

Never married 1.00 1.00

Married 1.20 (1.02-1.41) 0.026 1.65 (1.39-1.96) <0.001

Widowed 1.01 (0.78-1.34) 0.89 1.50 (1.13-1.99) 0.005

Divorced 1.08 (0.90-1.29) 0.40 1.50 (1.24-1.81) <0.001

Author’s response: It is not crude, it is adjusted estimates.

Discussion

19. Start the discussion with a summary of the key results of your study and then move on to discuss them in light of other relevant literature.

20. The findings from this paper may be helpful to your discussion and conclusion. Consider citing it as well.

Ng'ambi W, Mangal T, Phillips A, Colbourn T, Mfutso-Bengo J, Revill P, Hallett TB. Factors associated with healthcare seeking behaviour for children in Malawi: 2016. Trop Med Int Health. 2020 Dec;25(12):1486-1495. doi: 10.1111/tmi.13499. Epub 2020 Oct 19. PMID: 32981174.

21. Don’t repeat results in the discussion but rather compare your key results with those of other studies and explain any disparity or agreement.

Conclusion

22. This section should provide policy insights as well as alignment of the study findings with the SDGs as well as universal health coverage.

23. Include the policy relevance of your study findings as well.

 Author’s response: We have included policy relevance of our findings in the conclusion section.

---

## [Decision Letter · Decision Letter 3]

16 Feb 2022

Health care seeking behavior for common childhood illnesses in Birendranagar municipality, Surkhet, Nepal

PONE-D-20-18110R3

Dear Dr. Tiwari,

We’re pleased to inform you that your manuscript has been judged scientifically suitable for publication and will be formally accepted for publication once it meets all outstanding technical requirements.

Kind regards,

Filiberto Toledano-Toledano, Ph.D.

Academic Editor

PLOS ONE

Additional Editor Comments (optional):

Reviewers' comments:

Reviewer's Responses to Questions

**Comments to the Author**

1. If the authors have adequately addressed your comments raised in a previous round of review and you feel that this manuscript is now acceptable for publication, you may indicate that here to bypass the “Comments to the Author” section, enter your conflict of interest statement in the “Confidential to Editor” section, and submit your "Accept" recommendation.

Reviewer #4: (No Response)

2. Is the manuscript technically sound, and do the data support the conclusions?

Reviewer #4: Yes

3. Has the statistical analysis been performed appropriately and rigorously? 

Reviewer #4: Yes

4. Have the authors made all data underlying the findings in their manuscript fully available?

Reviewer #4: Yes

5. Is the manuscript presented in an intelligible fashion and written in standard English?

Reviewer #4: Yes

6. Review Comments to the Author

Reviewer #4: 1. They need to round the AOR to two decimal places.

2. AOR should be put in full as it is appearing for the first time.

3. The study variables should be put in a box. The current presentation does not make them look sexy.

4. Illnesses interact and their co-existence affects the HSB, may you refer to the paper by Ng’ambi et al (https://onlinelibrary.wiley.com/doi/epdf/10.1111/tmi.13499) and this could be another nice reference for your work.

5. The tables need proper formatting (refer to https://onlinelibrary.wiley.com/doi/epdf/10.1111/tmi.13499)

6. The figures need to have the titles.

7. It is not clear how the variables were selected into the model.

8. One limitation of the paper is that they only looked at two illnesses. The authors should check (https://onlinelibrary.wiley.com/doi/epdf/10.1111/tmi.13499) where a multiplicity of illnesses have been looked at.

7. PLOS authors have the option to publish the peer review history of their article (what does this mean?). If published, this will include your full peer review and any attached files.

Reviewer #4: **Yes: **Wingston Ng’ambi, Research Scientist- University of Geneva; Lecturer at Kamuzu University of Health Sciences

---

## [Editor Report · Acceptance letter]

21 Mar 2022

PONE-D-20-18110R3 

Health care seeking behavior for common childhood illnesses in Birendranagar municipality, Surkhet, Nepal: 2018 

Dear Dr. Tiwari:

I'm pleased to inform you that your manuscript has been deemed suitable for publication in PLOS ONE. Congratulations! Your manuscript is now with our production department. 

Kind regards, 

on behalf of

Dr. Filiberto Toledano-Toledano 

Academic Editor

PLOS ONE